

# Musculoskeletal simulations to examine the effects of accentuated eccentric loading (AEL) on jump height

Eric Yung-Sheng Su[1], Timothy J. Carroll[1], Dominic J. Farris[1,2] and Glen A. Lichtwark[1]

[1] School of Human Movement and Nutrition Sciences, Faculty of Health and Behavioural Sciences, The University of Queensland, Brisbane, Australia
[2] Sport and Health Sciences, College of Life and Environmental Sciences, University of Exeter, Exeter, Devon, United Kingdom

Corresponding author
Eric Yung-Sheng Su,
yungsheng.su@uq.edu.au

## ABSTRACT

**Background**. During counter movement jumps, adding weight in the eccentric phase and then suddenly releasing this weight during the concentric phase, known as accentuated eccentric loading (AEL), has been suggested to immediately improve jumping performance. The level of evidence for the positive effects of AEL remains weak, with conflicting evidence over the effectiveness in enhancing performance. Therefore, we proposed to theoretically explore the influence of implementing AEL during constrained vertical jumping using computer modelling and simulation and examined whether the proposed mechanism of enhanced power, increased elastic energy storage and return, could enhance work and power.

**Methods**. We used a simplified model, consisting of a ball-shaped body (head, arm, and trunk), two lower limb segments (thigh and shank), and four muscles, to simulate the mechanisms of AEL. We adjusted the key activation parameters of the muscles to influence the performance outcome of the model. Numerical optimization was applied to search the optimal solution for the model. We implemented AEL and non-AEL conditions in the model to compare the simulated data between conditions.

**Results**. Our model predicted that the optimal jumping performance was achieved when the model utilized the whole joint range. However, there was no difference in jumping performance in AEL and non-AEL conditions because the model began its push-off at the similar state (posture, fiber length, fiber velocity, fiber force, tendon length, and the same activation level). Therefore, the optimal solution predicted by the model was primarily driven by intrinsic muscle dynamics (force-length-velocity relationship), and this coupled with the similar model state at the start of the push-off, resulting in similar push-off performance across all conditions. There was also no evidence of additional tendon-loading effect in AEL conditions compared to non-AEL condition.

**Discussion**. Our simplified simulations did not show improved jump performance with AEL, contrasting with experimental studies. The reduced model demonstrates that increased energy storage from the additional mass alone is not sufficient to induce increased performance and that other factors like differences in activation strategies or movement paths are more likely to contribute to enhanced performance.

## INTRODUCTION

During explosive movements, such as jumping and throwing, humans typically utilize stretch-shortening cycles (SSC) by first performing an eccentric loading that aims to increase the force and power in the subsequent concentric movement. Numerous studies on isolated muscles (*Cavagna & Citterio, 1974*; *Cavagna, Dusman & Margaria, 1968*; *Cavagna, Saibene & Margaria, 1965*) and *in vivo* human experiments (*Bobbert et al., 1996*; *Cronin, McNair & Marshall, 2001*; *McBride, McCaulley & Cormie, 2008*; *Sheppard, Newton & McGuigan, 2007*) have confirmed that SSC can effectively increase the concentric force and power output. However, the movement dynamics need to be precisely tuned so that interaction between elastic and contractile elements in muscles enables maximization of power output during the concentric phase of the movement (*Ishikawa, Finni & Komi, 2003*; *Ishikawa et al., 2006*).

Accentuated eccentric loading (AEL) is a form of movement manipulation that has been suggested to enhance power output. AEL is a type of SSC that requires the person to perform a heavy eccentric loading (added mass or force) followed by a light concentric loading. During the AEL movement, the added external load is released at the transition from the eccentric to the concentric phase. Some studies found that the acute response to AEL could increase jump height by 4.3~9.52%, peak power output by 9.4~23.21%, and maximal concentric vertical ground reaction force by 3.9~6.34% during a countermovement jump (*Aboodarda et al., 2013*; *Sheppard, Newton & McGuigan, 2007*). Similarly in bench press, *Doan et al. (2002)* reported an increased concentric force and *Ojasto & Häkkinen (2009)* found increased concentric power. By contrast, *Aboodarda et al. (2014)* reported that AEL applied through elastic resistance during a drop jump did not alter jump height, muscle activation level, or other kinetics profiles during the concentric push-off phase. Indeed, a review study by *Wagle et al. (2017)* concluded that current evidence for both acute responses and chronic adaptation to AEL is inconsistent, possibly due to different exercises selected, training equipment used, or the load selected across different experiments. As a result, more research is needed to clarify the effects of AEL on force, work, and power production during explosive SSC movements.

There are a number of potential mechanisms that might drive enhanced power output during AEL movements. The most common explanation for why AEL should enhance power is that increased load amplifies elastic energy storage in the tendon and aponeurosis, which can then be released in the concentric phase (*Wiesinger et al., 2017*). For instance, AEL countermovement jump may result in greater force generation in the descent to decelerate added inertia, potentially resulting in greater tendon loading prior to the upward motion. However, these effects on tendon loading or storage and return of energy are yet to be tested. At the whole-body level, one potential mechanism for enhanced performance is that AEL increases the net vertical impulse during the ground contact phase. This would increase take-off velocity and hence jump height. *Bobbert et al. (1996)*

compared squat jump and CMJ performance and argued that CMJ conditions increased the ground contact time to build up a higher muscle force, which helped to increase the net vertical impulse, take-off velocity, and hence jump height. During the initial descent of the CMJ, the negative vertical impulse must be greater in AEL CMJ than in non-AEL CMJ because the overall mass is greater with AEL. However, to slow down the descent, the positive vertical impulse that is generated to decelerate the greater mass must also be greater in AEL CMJ than in non-AEL CMJ. When we examine the whole descent phase, the net vertical impulse must be zero at the bottom of the countermovement regardless of the conditions. Therefore, what is unknown is whether the additional positive impulse generated during the push-off phase can be greater in AEL CMJ than in non-AEL CMJ. Whilst we might expect that additional mass prior to release will increase force and hence vertical impulse in the upward phase, this effect has yet to be examined.

Musculoskeletal modelling and simulation provide a powerful way to understand how AEL might enhance power output. Simulation studies have suggested that the application of an external load can result in MTU power amplification in some cases (*Galantis & Woledge, 2003*; *Sawicki, Sheppard & Roberts, 2015*). For example, when an intermediate external load was applied to a Hill-type frog MTU model during stretch-shortening cycles or *via* a simulated catch mechanism, MTU power was amplified above the maximal fibre power predicted from the muscle's force-length-velocity relationship (*Richards & Sawicki, 2012*; *Sawicki, Sheppard & Roberts, 2015*). The extra MTU power was attributed to energy storage from the tendon. However, this effect was highly sensitive to the effective mass, such that only a narrow optimal range of external load magnitudes produced power amplification (*Richards & Sawicki, 2012*). Multi-segment musculoskeletal simulations have also found that adding or removing weight impacts performance of maximal effort jumps (*Bobbert, 2014*), however, the effect was smaller than that reported in similar experimental studies (*Markovic & Jaric, 2007*; *McBride, Haines & Kirby, 2011*; *Pazin et al., 2013*). These simulation studies show that applying additional load to the body can impact maximal task performance, and the magnitude of the mass or resistance might be critical to allow this performance enhancement to occur. Among the experimental studies that showed enhanced AEL jumping performance, the most common range of AEL loads was 15–30% body mass (*Aboodarda, Page & Behm, 2015*; *Aboodarda et al., 2013*; *Sheppard, Newton & McGuigan, 2007*). To date, however, there have been no simulation studies exploring how sudden manipulation of loads applied during a movement, as per AEL protocols, might enhance work or power output from the musculoskeletal system.

The primary aim of this study was to use simulations to explore how AEL influences muscle work and power during a jumping task. We chose a simplified model to examine whether additional load during the eccentric phase of movement enhances muscle performance *via* elastic tendon-loading mechanics, without confounding effects of changes in body posture, joint coordination and range of motion. Our simplified biomimetic mechanical system constrained movement of the trunk (vertical movement only), and consisted of only two segments (representing the tight and shank) and was powered by four muscles acting around the hip and knee joints. The AEL conditions tested in our simulations were 15% and 30% additional body mass. We hypothesised that AEL
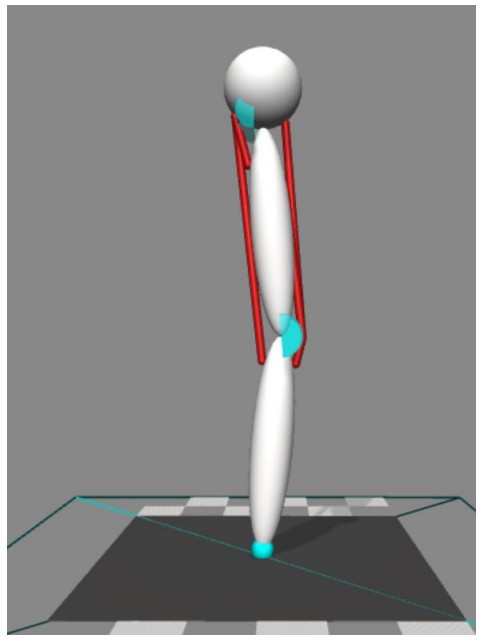

**Figure 1  Four muscles model to perform two-dimensional forward dynamic simulation.**

simulations would enhance work and power compared to simulations in which mass did not change.

## MATERIALS & METHODS

### Model details

Our simulations were performed using OpenSim 4.0 software (*Delp et al., 2007*; *Seth et al., 2018*). We performed two-dimensional forward dynamic simulations (Fig. 1). The model consisted of a body representing the head, arms, and trunk (HAT) and two rigid segments representing the thigh and shank. The HAT segment was constrained so that it only moved in the vertical direction with no rotational degree-of-freedom. HAT, thigh, and shank segments were connected by hinge joints representing the hip and knee. The joints were only allowed to move freely within a specified range (hip = 0°∼90° hip flexion, knee = 0°∼135° knee flexion, zero represents the joint angle at upright posture, positive value represents the direction of joint flexion). Joint range was constrained by applying a spring-damper model that applied a resistive torque in proportion to the change of angular velocity beyond the prescribed range. This spring-damper model had an equivalent function to the passive anatomical structures that stabilize the joint (*i.e.,* ligaments and joint capsules), and the parameters in the spring-damper model are provided in Table 1. The model's mass, inertial properties and segment lengths (Table 1) were adapted from the 10-segment, 23 degree-of-freedom model developed by *Anderson & Pandy (1999)*. The mass and inertial properties of HAT include all segments from pelvis and above. The mass of the thigh and shank segments were twice those in *Anderson & Pandy (1999)*, in

**Table 1   Model parameters.**

| Segmental properties of the model's segments | | | |
|---|---|---|---|
| | Mass (kg) | Moment of inertia (kg m$^2$) | Diameter (m) or length (m) |
| Ball-shaped Body (HAT) | 47.64 | 1.482 | 0.15 |
| Thigh Segment | 19.44 | 0.146 | 0.4 |
| Shank Segment | 7.68 | 0.053 | 0.4 |
| **Hunt-Crossley parameters** | | **Spring-damper parameters** | |
| Stiffness (N/m) | 10$^7$ | Upper Stiffness (N.m/degree) | 10 |
| Dissipation | 20 | Lower Stiffness (N.m/degree) | 1 |
| Static coefficient of friction | 2 | Transition (degree) | 3.0 |
| | | Damping (N.m/degree/s) | 100 |

order to represent two legs combined. The mass and inertial properties were all scaled to one subject (height: 1.78 m, mass: 78 kg), and the segment lengths were also taken from the same subject. We added a Hunt-Crossley contact model to define the contact points between the model and ground (*Hunt & Crossley, 1975*), with values tuned to ensure that there was no premature take-off of the model during descent and limited sliding of contact spheres. Parameters are provided in Table 1.

## Muscle actuators

We used four different muscle–tendon-unit (MTU) actuators to represent the major muscles in the lower limb. These MTU actuators were based on a three element Hill-type muscle model (*Zajac, 1989*). These MTU actuators were the vasti muscle groups (VAS), rectus femoris (REC), gluteus maximus (GLU), hamstring muscle groups (HAM). The Opensim muscle model used in our simulation was the Millard2012EquilibriumMuscle model (*Millard et al., 2013*). The input to the model was muscle excitation, which represented the excitatory signal from the peripheral nervous system that activated the muscle. Muscle excitation ranged between 0 and 1, which also led to muscle activation in the model *via* a first-order differential equation representing excitation-contraction coupling (*Millard et al., 2013*).

## Muscle parameters

The muscle parameters were adapted from the leg muscle model by *Delp et al. (1990)*. Maximal isometric force ($F_{max}$) of each muscle was scaled to the power of 2/3 relative to the scale factor applied to mass and inertial scaling. This was based on the scaling relationship between segment mass and segment cross-sectional area, assuming uniform segment density. $F_{max}$ of each muscle was then doubled so that it matched the force generated by both legs. Our muscle model assumed no pennation angle for simplicity. We modified $L_{opt}$ and $L_{tendon}$ so that the passive tension from the muscle did not exceed 5% of the $F_{max}$ value during the model's constrained jumping motion. The muscle parameters are provided in Table 2.
| Table 2 | Muscle parameters. | | |
|---|---|---|---|
| | $F_{max}$ (N) | $L_{opt}$ (m) | $L_{tendon}$ (m) |
| VAS | 9272 | 0.1765 | 0.2065 |
| REC | 2393 | 0.1766 | 0.3144 |
| GLU | 3888 | 0.0671 | 0.0679 |
| HAM | 6796 | 0.1166 | 0.3584 |

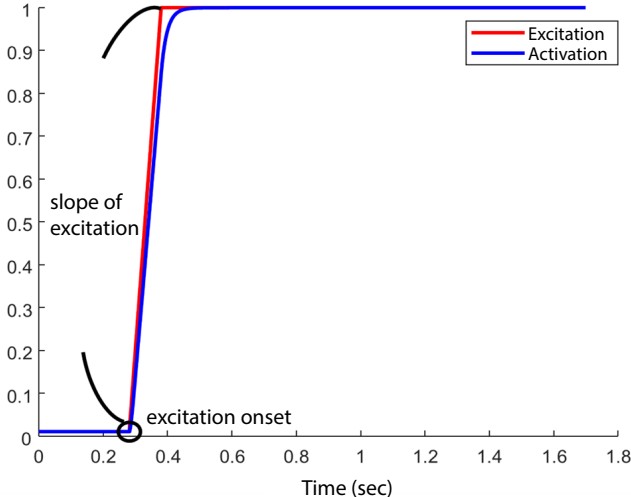

**Figure 2** Example of one combination of " excitation onset" and "slope of excitation."

## Model excitation

At the start of each simulation, the model was set at an initial posture at 5° hip flexion, 10° knee flexion such that the HAT segment would fall, and the knee and hip joints would flex, due to gravity. All muscles began the simulation with the minimum muscle excitation of 0.01. The initial muscle activation was set to 0.01 for all muscles. We did not specify the initial fibre length, instead Opensim computed fibre length based on the initial muscle activation and other muscle parameters detailed in Table 2 (equilibration occurring at start of simulation). Muscle excitation remained at 0.01 until the *excitation onset* for each muscle, after which time the excitation increased to a maximum value of 1 with an excitation rate or *slope of excitation* between 1/s and 10/s. Therefore, the model's motion was determined by the interaction between the *excitation onset* and *slope of excitation* (Fig. 2) of each muscle and the passive mechanics of the model. An optimization (details below) was run to determine the optimal combination of muscle excitation onsets and slope of excitations to achieve the maximum jump height in each model. We set the duration of the simulation to 1.7 s to ensure sufficient time for the model to reach the highest point in the airborne phase.

## Cost function and optimization routine

The primary cost function was based on maximizing jump height, however there was also a penalization term. To prevent the model utilizing the spring properties set to mimic passive joint structures at the end of range for each joint, we penalised such motion (assuming excessive flexion/extension is not desirable due to the likelihood of injury). In each simulation, the joint limit for the knee (135° knee flexion) at the bottom of the squat was used as the kinematic penalty term in the cost function. Given the geometry of the model, the hip joint would never reach its joint range throughout the jumping motion until take-off, and therefore was not included in the penalty term.

The optimization criteria were: (1) maximize jump height, (2) minimize knee joint limit penalization term. The weights for each parameter were determined to ensure that minimum spring-damper torque was applied during the movement, whilst ensuring jump height was prioritised. The weights were hand-tuned with arbitrary values on a trial-and-error basis. Jump height was defined as the difference in HAT centre of mass position between the model's initial posture and the highest position achieved in the simulation. The optimization cost function (J) is provided below, where H represents jump height and $\theta$ represents maximal knee angle in the jump:

$$J = H, \text{if } \theta < 135° \tag{1.1}$$

$$J = H - 0.01\left(\theta - 135°\right), \text{otherwise.} \tag{1.2}$$

The numerical optimization was performed in MATLAB (version R2018b; MathWorks, Natick, United States). We used a nonlinear simplex algorithm "fminsearchbnd" (D'Errico, 2020) to search the optimal solution for our cost function within bounds given above. "fminsearchbnd" is an extension of MATLAB "fminsearch" algorithm, but it also accommodates the parameter bounds of all input variables when searching for global minimum. To find the global maximum of our cost function, we multiplied our cost function by $-1$ so that the minimum value from "fminsearchbnd" represented the maximum value of the cost function.

There were three conditions to be optimized: the normal condition, a 15% AEL condition, and a 30% AEL condition. In the normal condition, the model mass was kept at the original mass (Table 1) to represent a standard countermovement jump without external load. In AEL conditions, there were two steps involved in the optimization process (termed 'split' method). In step 1, we performed an optimization search of the cost function provided above with 15% and 30% additional body mass added to the model (added to HAT segment), and the parameters to be optimized were muscle excitation onsets and slope of excitations. After finding the optimal solution for each added mass, step 2 involved starting a new simulation from the beginning of upward motion with the added mass removed from the model to simulate the AEL concentric condition. The exact states [muscle excitations, muscle fibre and tendon lengths and joint position and velocity] at the beginning of upward motion in step 1 were implemented at the start of the simulation of step 2. A new optimisation of muscle excitation parameters during step 2 was conducted, as some muscles may not have been excited during the optimisation

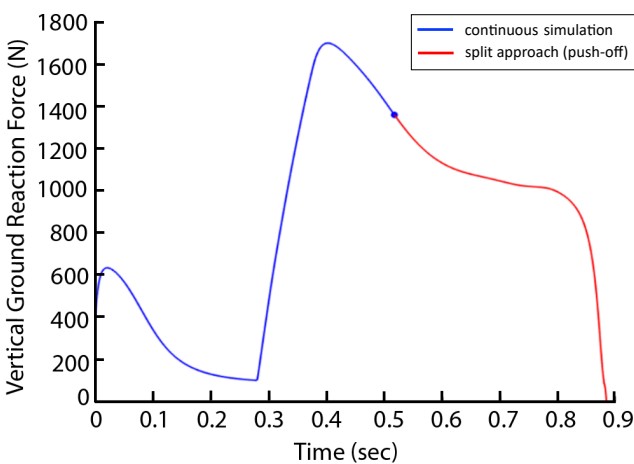

**Figure 3 Vertical ground reaction force during the non-AEL condition using the original (continuous simulation) and split approaches.** The graph shows the results from the pilot simulation using a single muscle (knee extensor) model. The *x*-axis shows the time (second) from the start of the descent to take-off, and the *y*-axis shows the vertical ground reaction force (N). The blue line represents the original approach (continuous simulation), and the red line represents the push-off phase from the split approach.

in step 1. We approximated the beginning of upward motion as the time when the VAS fiber velocity became zero and hence started generating positive work. Originally, we used the lowest COM height to determine the turning point of the simulation. However, to reduce discontinuities in muscle forces that occurred across the split simulations, we instead approximated this turning point *via* VAS fiber kinematics. We found that the time when fiber velocity approached zero occurred at a very similar time to the model's turning point (0.0201 and 0.0206 s difference for 15% and 30% AEL conditions), with limited discontinuity. We also tested our simulation "split" method (two-step approach) on the non-AEL condition in our pilot simulations with one muscle model. We found very similar results between split-method and non-split (original) approach. Take VGRF for example, the curves perfectly aligned (Fig. 3). We are confident that using split method in the AEL conditions was valid for this study. We then compared the muscle and model dynamics during upward motion (*i.e.,* push-off phase) in three different conditions.

We used a "split" method because we were unable to perform a simulation that changed the model's mass halfway through the movement in the Opensim platform. Whilst it is possible to add an external force in the descent phase of movement and remove this external force during ascent, this solution does not account for the effect of inertia during the descent. The "split" method assumes that the optimal AEL jump (added mass removed at the turning point) shares the same eccentric portion of the optimal jump with added mass (executed to completion). While this assumption may not be true of jumping humans, who may select different squat depths or rates of descent between AEL and added mass jumps, our purpose here was to exclude effects like change of squat depth so that we could examine the fundamental fibre-tendon mechanics differences between AEL and non-AEL conditions in isolation. The maximum jump achieved under all weight conditions always

occurred for the deepest squat possible, where force is applied through the maximum range. As such it seems that this is a fair comparison of optimal jumping across conditions.

## Simulation data analysis

Squat depth was defined as the difference in HAT centre of mass position between the model's initial posture and the lowest position achieved in the simulation. Muscle power was calculated as muscle work divided by push-off time, and therefore was the average power during push-off. The push-off time was defined as the time for the HAT centre of mass to move from the beginning of upward motion until take-off.

The total muscle work in the model did not necessarily equate to the effective vertical work done on the whole system. This is because some ineffective energy was expended during the jumping motion, such as the horizontal and rotational kinetic energy of each segment. To understand how AEL affected the whole system dynamics, the model's vertical ground reaction force (VGRF), center of mass (COM) vertical velocity, and COM vertical power were calculated. The vertical component of the Hunt-Crossley contact force between the model and the ground was taken as the VGRF. The COM vertical power was calculated as the product of the VGRF and the COM vertical velocity, and only the push-off phase was analyzed.

## RESULTS

### Squat depth

The AEL conditions achieved a similar squat depth to the normal condition (less than 0.5% difference in squat depth). Consequently, the joint ranges produced by the model did not differ meaningfully across conditions, which means that any performance difference found in different conditions was not simply caused by the movement range.

### Muscle work and power

The muscle work and power contributions during the push-off phase are provided in Fig. 4. The individual and total (*i.e.,* sum of all muscles) muscle works and powers are compared across normal, 15% AEL, and 30% AEL conditions as the percentage difference relative to the normal (no AEL) condition. Both high and low AEL conditions showed a negligible change in total muscle work (less than 1% reduction) compared to the normal condition. Since our model achieved similar squat depths across conditions, the total muscle work during the push-off phase primarily determined the effective jump height in our model. We found a negligible change in effective jump height (less than 1% reduction) in both AEL conditions compared to the normal condition (Table 3). The major finding was that AEL did not increase total muscle work or effective jump height in our model.

Figure 4 shows an overall small increase in total muscle power in AEL conditions compared to the normal condition, with the percentage increase slightly higher in the 15% AEL condition compared to the 30% AEL condition. Nevertheless, the percentage increase was less than 3%, which might be considered a relatively small effect for average muscle power, noting that this did not cause an increase in jump height.

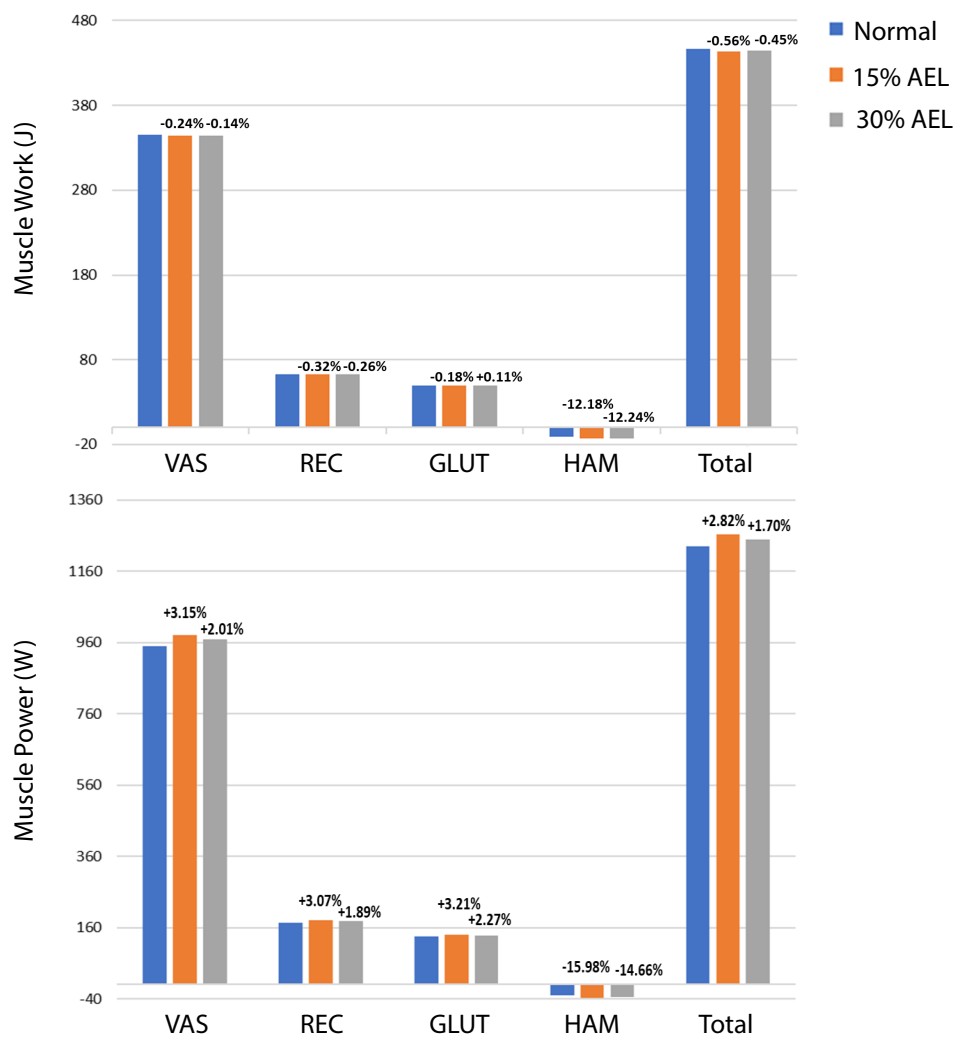

**Figure 4** **Muscle work and power during push-off phase across normal, 15% AEL, and 30% AEL conditions.** The percentage values above bars denotes the relative change from the corresponding normal condition. Positive sign represents increased work and power, and negative sign represents decreased work and power.

## Whole system dynamics

The relative differences in peak VGRF and peak COM vertical power during push-off between AEL conditions and the normal condition are summarized in Table 3. Our model showed negligible changes (less than 1% reduction) in peak VGRF and peak COM vertical power. Furthermore, the VGRF and COM vertical power curves relative to time are very similar in shape (Fig. 5), which also explains why our model had negligible change in effective jump height (less than 1%).

## Dynamics of muscle

Given that the jumping motion was primarily knee-dominant, we use the results from the VAS muscle–tendon dynamics to explore why there was a lack of AEL improvement in

**Table 3** **Percentage difference in effective jump height, peak VGRF and peak COM vertical power during push-off between AEL conditions and normal condition (chosen as baseline).** Positive/negative sign denotes the increase/decrease in value compared to baseline.

| | Condition | Percentage difference |
|---|---|---|
| Effective jump height | Normal | baseline |
| | 15% AEL | −0.798% |
| | 30% AEL | −0.899% |
| Peak VGRF (N) | Normal | baseline |
| | 15% AEL | −0.16% |
| | 30% AEL | −0.12% |
| Peak COM vertical power (W) | Normal | baseline |
| | 15% AEL | −0.13% |
| | 30% AEL | −0.07% |

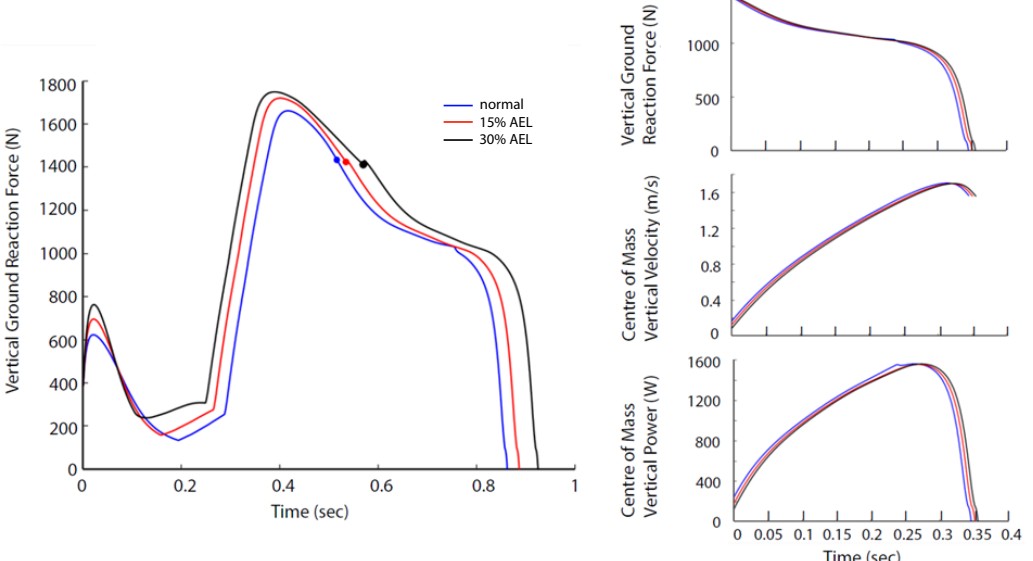

**Figure 5** **Force, velocity, and power profile representing the whole system dynamics in both models across normal, 15% AEL, and 30% AEL conditions.** The graph in the left column shows the VGRF for the entire jumping motion until take-off. The color-filled markers represents the time when the model achieved the lowest posture. Three different conditions in each model were time-normalized from the lowest posture to take-off. The three graphs in the right column show the time-normalized VGRF, COM vertical velocity, and COM vertical power during the push-off phase.

muscle work, muscle power, and the whole system dynamics. Figure 6 shows the activation, MTU force, tendon length, fiber length, and fiber velocity for VAS for the entire jumping motion (optimized) across three different loading conditions. The 30% AEL condition showed the earliest increase in VAS activation, followed by the 15% AEL condition, and then the normal condition. This allowed the 30% AEL condition to produce the required active muscle force in the descending phase with sufficient time to decelerate the system's

COM before the end of joint range (which would otherwise be penalized in the optimization cost function). As a consequence, VAS MTU force was higher than in the normal condition during the descending phase in both AEL conditions (Fig. 6).

Prior to achieving the model's lowest position, VAS was already at full activation across all three conditions (Fig. 6). Peak activation occurred at approximately the middle of eccentric phase, and force then decreased until the lowest point in the movement such that the VAS MTU force at the start of the push-off differed by less than 0.1% between AEL and normal conditions. The VAS tendon length (less than 0.001% difference), fiber length and fibre velocity (less than 0.05% difference) also had similar magnitudes across three different conditions at the start of the push-off (Fig. 6). Therefore, this analysis shows that each condition resulted in very similar states at the start of the push-off phase, regardless of added mass in the AEL conditions. The model in AEL conditions therefore behaved similarly to the normal condition during push-off.

## DISCUSSION

This simulation study explored the mechanisms of putative work and power enhancement due to AEL during jumping. Our major findings contradicted our original hypothesis. We found that neither AEL load condition increased total muscle work or effective jump height during an optimal constrained countermovement jump. Our simulations actually showed slight reductions in performance (less than 1%), which differed from the *in vivo* studies that reported 4.3~9.52% increases in effective jump height by utilizing AEL (*Aboodarda et al., 2013*; *Sheppard, Newton & McGuigan, 2007*). We also found that our model had negligible difference (less than 1%) in peak VGRF and peak COM vertical power during push-off across conditions, contrasting with *in vivo* studies that reported 3.9~6.34% increases in maximal concentric vertical ground reaction force, and 9.4~23.21% increases in peak concentric power output by utilizing AEL (*Aboodarda et al., 2013*; *Sheppard, Newton & McGuigan, 2007*).

We also found that the agonist muscles (VAS and REC) generated slightly less muscle work in the AEL conditions, which contributed to slightly less COM work and jump height. However, the push-off duration was also shorter in the AEL conditions, contributing to a slightly higher muscle power despite a minimal reduction in muscle work. The changes in jump height and muscle power in AEL conditions were minimal, and are best interpreted as being no different between conditions.

The results indicate that the lack of difference in jump performance was a result of the states of the system being equivalent at the bottom of the movement, despite differences in the weight of the model/system at this point. Our results showed that our model descended to a similar squat depth to ensure the whole joint range was fully utilized in the push-off phase, regardless of the conditions. Thus, our model began push-off at a similar posture, similar fiber length and fiber velocity, similar tendon length, similar fiber force, and the same (*i.e.,* complete) activation level. Therefore, the optimal solution predicted by our model was primarily driven by intrinsic muscle dynamics (force-length-velocity relationship), and this coupled with the similar model state at the start of the push-off led

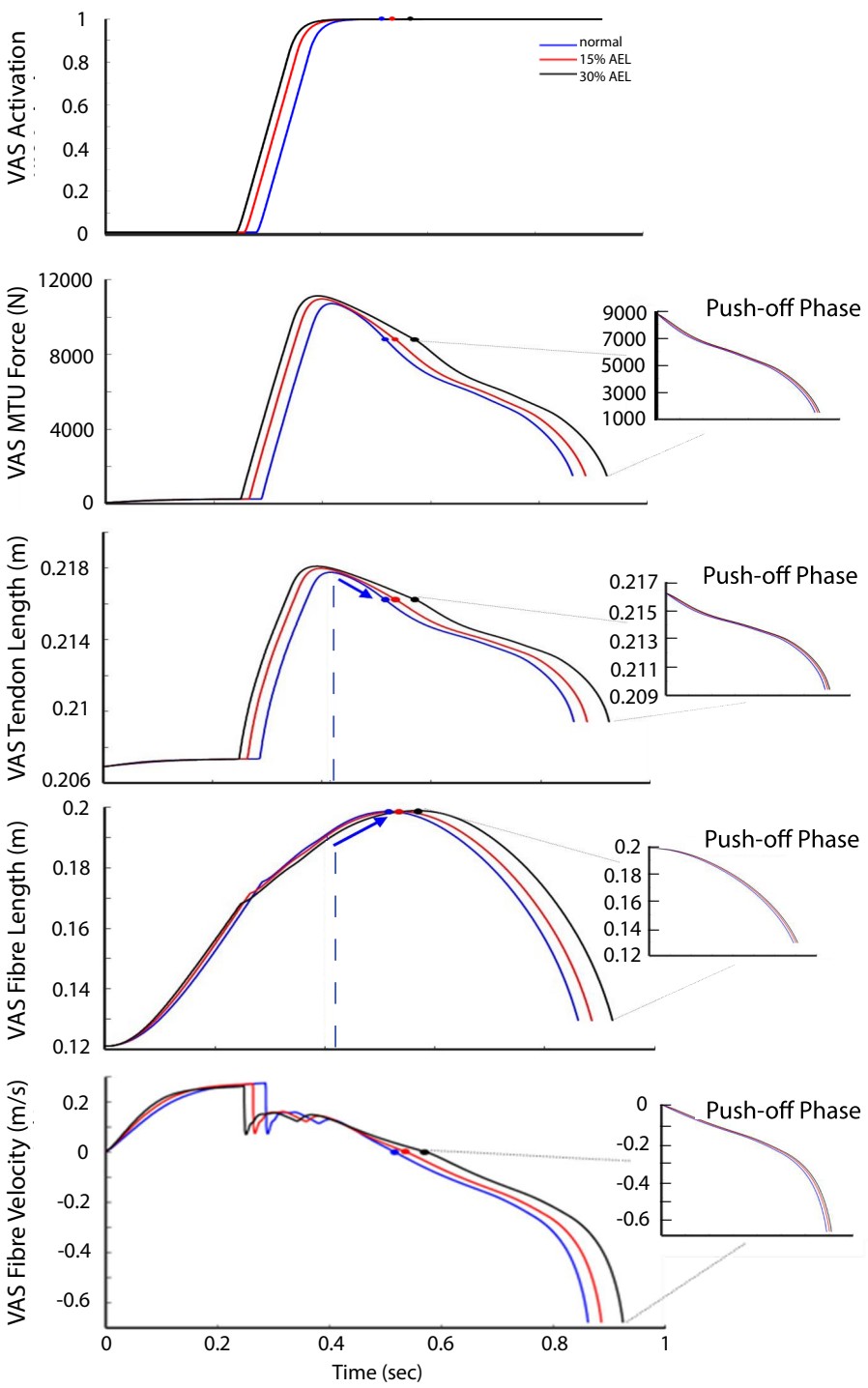

**Figure 6** **VAS activation, MTU force, tendon length, fiber length, and fiber velocity across normal, 15% AEL, and 30% AEL conditions.** The main graphs show the data for the entire jumping motion until take-off. The color-filled markers represent the time at the beginning of upward motion for each condition. The dashed blue lines represent the time when the tendon achieved its maximal length in the normal condition. The blue arrows indicate the change in length in the normal condition. Three different conditions were time-normalized from the beginning of upward motion to take-off (*i.e.*, push-off phase), as shown in the smaller inset plots.

to similar push-off performance across all conditions. The reason the states were equivalent at the release point of the weights (beginning of push-off) seems to be that the optimal solution required that the muscle achieved its full activation prior to the bottom of the movement to utilize maximum muscle and tendon work on the ascent. As such, at the bottom of the movement (with the same joint angle across conditions), the maximum force was dictated by the maximum force capability of a maximally activated muscle, and this was equivalent in different conditions.

One explanation that has been proposed as a potential mechanism for enhanced performance in AEL is storage of elastic energy in tendons. In this regard, it is important to remember that tendon length is a function of MTU force. Our simulations showed that AEL conditions caused a larger tendon excursion during the descending phase (Fig. 6). However, the additional elastic energy stored in the tendon was already returned to the system before the model achieved its lowest posture. This returned elastic energy from the tendon was absorbed into the contractile element (or muscle fibers) prior to achieving the lowest position, demonstrated by the shortening of the VAS tendon (as force declined) while the MTU continued to lengthen *via* fibre (contractile) lengthening (Fig. 6). As a result, there was no difference in the stored elastic energy between different conditions when compared at the bottom of the movement. In other words, our constrained jumping model predicted the same available elastic energy to be returned into the push-off phase, and therefore our proposed tendon-loading mechanism did not facilitate an increase in elastic energy in AEL conditions compared to normal conditions. The reason our simulations found this unique movement strategy was because the optimizer only searched the movement solution producing the highest jump height, and this unique strategy was the best solution given the constraints of the model. Storing elastic energy in the tendon was not the criteria of the optimization process, and therefore the optimizer did not account for this factor.

There might be concern over whether the model gave sufficient validity to explore the AEL jumping movements because of the simplicity of the model. To address this major concern, we compared our simulated jump height results to the ranges measured in the literatures that included trunk constraints during human jumping (*Kubo et al., 2007*; *Pérez-Castilla et al., 2020*), which were reported to be around 20 cm. Considering that our model removed the foot segment and ankle joint rotation, the values from our model (around 10 cm) was a good approximation of the knee-dominant motion. We also examined the general timing of muscle activation during human jumping (*Held, Siebert & Donath, 2020*; *Padulo et al., 2013*) and found that vastus EMG achieved close to maximum value at the bottom of the jump, and the biceps femoris EMG achieved maximum value after take-off. These EMG activation patterns were similar to the results of our model. Considering the fact that the model found the deepest squat depth as the optimal solution, this demonstrates our model operates in a similar (albeit abstract) way to the human body during jumping (*Sánchez-Sixto, Harrison & Floría, 2018*).

There are some limitations to interpreting the findings based on our modelling assumptions. Firstly, our model assumed a 'bang–bang' muscle control towards maximal excitation, and therefore the muscles were not allowed to use submaximal activation after the excitation slope finished ramping up. Muscle control in the human musculoskeletal

system is often more complex than that described by a simple bang–bang assumption. EMG characteristics have been shown not to differ between AEL and body-weight drop jumps during push-off (*Aboodarda et al., 2014*), however not all muscles are likely to be fully active prior to the upward propulsion phase of jumping (*Bobbert et al., 1996*), in contrast to the simulations specified by our optimal solution. In reality, different muscles may achieve different activation levels over time during human jumping and the bang–bang muscle control used in the simulation may have limited our model's representativeness, because it did not accurately simulate a human-like muscle activation. However, all optimal solutions in our simulation specified maximum muscle activation prior to upward movement and hence it is likely that this is the optimal method in our simplified model, regardless of how the muscle gets to this activation level. In another jumping simulation study with more complicated muscle control (*i.e.,* step function), the mono-articular muscles still produced the similar activation pattern while some bi-articular muscles behaved differently (*i.e.,* REC activation ramped up after push-off began) (*Nagano, Komura & Fukashiro, 2007*). Our simulations also found the lowest squat depth to be the optimal solution, whereas real humans do not necessarily utilize the deepest squat depth when performing a self-selected maximal jump (*Mandic, Jakovljevic & Jaric, 2015*). Although previous simulation studies have also shown that increasing squat depth should improve maximal jump height (*Bobbert et al., 2008*; *Domire & Challis, 2007*), this relationship has not been observed in human experiments in which healthy adults and elite athletes participated (*Domire & Challis, 2007*; *Mandic, Jakovljevic & Jaric, 2015*). Possible explanations to avoid theoretically optimal deeper squat depths include joint discomfort and joint protective mechanisms. Another physiological property of the muscle not considered in our model is residual force enhancement (*Hahn et al., 2010*). Residual force enhancement increases the muscle force after the lengthening of the muscle fibre (*Hahn et al., 2010*). As a result, stretch-shortening cycle (SSC) movements might benefit from residual force enhancement and hence power enhancement. Residual force enhancement was not simulated in our muscle model, adding another limitation to our study. However, when examining gross movement tasks, residual force enhancement has been found to have little or no effect on the magnitude of force production (*Brown & Loeb, 2000*). Therefore, residual force enhancement observed at the single muscle is unlikely to contribute significantly to the force enhancement during a multi-joint, multi-muscle SSC. Finally, we did not test the effect of adding a foot segment with multiple contact points, which allows the centre of pressure to translate. However, we expect that the results will be similar with more complex models, providing the optimal solution requires that muscles are maximally active prior to take-off.

It is important to consider why AEL may cause work and power enhancement in some human experiments (*Aboodarda et al., 2013*; *Sheppard, Newton & McGuigan, 2007*), but not in our simulations. *Sheppard, Newton & McGuigan (2007)* reported the same squat depth being selected by the participants with improved jump height under AEL conditions. Similarly, our model also predicted the same squat depth between the AEL and non-AEL conditions. Furthermore, our model was highly constrained so that there was also no difference in joint range between AEL and non-AEL conditions. Therefore, any difference in work, power, or jump height during the push-off phase can only be determined by

the force produced by the muscle in our model. *Sheppard, Newton & McGuigan (2007)* proposed that AEL may have caused a higher muscle force at the initiation of the upward velocity to cause work enhancement. However, our simulations predicted that the highest muscle force occurred before the model achieved the lowest position, and that the muscle force was similar at the initiation of the upward acceleration across different added load (AEL) conditions. We believe that the discrepancy between these *in-vivo* data and our simulation findings most likely occurred because our control scheme (bang–bang) is different to that employed biologically. It remains to be seen, however, why humans do not adopt the optimal solution that requires maximum muscle activation prior to upward movement. Potentially, activation is sub-optimal without AEL during actual CMJ, and AEL changes mechanics/activation patterns to improve jumping performance. It could be argued that our model is already maximising the storage and generation of energy (*i.e.,* full activation and same states at turning point), and hence AEL cannot improve performance further. An alternative view is that humans are able to find an optimal solution that our model cannot because humans achieve greater jump heights with AEL. Maybe, if our model could find the optimal control strategy that humans use, our solution would be able to make more use of energy stored in tendon. As such, we suspect that *in-vivo* findings of superior performance with AEL compared to normal jumping may be attributed to neural control factors, which might (or might not) produce a movement pattern that stores and returns more elastic energy in the tendon.

## CONCLUSIONS

In this simulation study, we found that countermovement jump performance (*i.e.,* jump height, COM vertical power) did not improve with AEL, irrespective of the magnitude of added load. This lack of effect primarily occurred because both AEL and non-AEL conditions had the same squat depth, muscle activation level, and muscle/tendon force at the start of the upward motion. Since AEL did not change model tendon strain at the start of the push-off phase, it did not affect the amount of stored elastic energy available for return in the push-off phase. Therefore, our results highlight that AEL does not take advantage of a potential tendon-loading effect to enhance work and power output in our optimised simulation. Our findings assumed that the major lower limb muscles already achieved and were able to maintain full activation throughout the push-off phase. However, utilizing bang–bang muscle control might have mis-represented human jumping control, and therefore our findings provide only theoretical evidence that altering mechanical loading in a simple but highly constrained musculoskeletal system does not affect effective work and power output. Changing the muscles' activation profiles might alter how elastic energy interacts within the jumping system; however, more research is still needed to explore this speculation.

## ACKNOWLEDGEMENTS

We gratefully thank Dr. Matthew Millard (Heidelberg University) in providing technical assistance and suggestions in developing our modelling and simulation protocols.

### Funding

Eric Yung-Sheng Su is the recipient of a UQ Graduate School Scholarship. Glen Lichtwark receives salary support via the Australian Research Council Future Fellowship (FT190100129) award. The funders had no role in study design, data collection and analysis, decision to publish, or preparation of the manuscript.

### Grant Disclosures

The following grant information was disclosed by the authors:
UQ Graduate School Scholarship.
Australian Research Council Future Fellowship: FT190100129.

### Competing Interests

The authors declare there are no competing interests.

### Author Contributions

- Eric Yung-Sheng Su conceived and designed the experiments, performed the experiments, analyzed the data, prepared figures and/or tables, authored or reviewed drafts of the article, and approved the final draft.
- Timothy J. Carroll conceived and designed the experiments, analyzed the data, authored or reviewed drafts of the article, and approved the final draft.
- Dominic J. Farris conceived and designed the experiments, analyzed the data, authored or reviewed drafts of the article, and approved the final draft.
- Glen A. Lichtwark conceived and designed the experiments, analyzed the data, authored or reviewed drafts of the article, and approved the final draft.

### Data Availability

The simulation codes, simulation animations, and results are available in the Supplemental Files.

### Supplemental Information

Supplemental information for this article can be found online at http://dx.doi.org/10.7717/peerj.14687#supplemental-information.

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
