# Peer review of "Musculoskeletal simulations to examine the effects of accentuated eccentric loading (AEL) on jump height"

_PeerJ, doi:10.7717/peerj.14687_

## Round 0.1 · original submission · Major Revisions

Two reviewers have given exhaustive, constructive and rigorous reviews; both are quite supportive of the study. More information, and potentially analysis, is needed on issues such as validation and simulation results and key assumptions. The paper will need re-review but holds plenty of promise. Thank you for submitting this interesting study to PeerJ.

Reviewer 1 ·

Basic reporting

no comment

Experimental design

no comment

Validity of the findings

no comment

Additional comments

(1) The study is a great example of a good use of computer modeling, attempting to isolate an effect that is difficult to isolate in experiments on live humans and with conflicting results on these experiments in the literature.

(2) My largest concern was the model's validity was not reported and this made it difficult for me to have confidence in its results on the AEL question. The authors seemed to already recognize this too, from the speculative "...due to the simplicity of our model" comment at end the Abstract. There were some simplifying assumptions in the model that deviate from real humans doing real jumping in fairly large ways, e.g. no trunk rotation, no foot, and no ankle plantarflexors. These are all really important elements of the body for jumping, which I think is why most simulation models of jumping are still fairly "complex" models, e.g. Bobbert studies. It would go a long way towards giving readers confidence in using the model for the AEL question if it can be shown that the model does "normal jumping" in a reasonably realistic way despite these simplifying assumptions, e.g. is the amount of flexion in the countermovement realistic, is the maximum jumping height realistic, is the timing of the muscle activity realistic? If the model doesn't do these things realistically then it seems questionable to move onto using it for the AEL question. My suggestion is to add an assessment of the model’s validity to the manuscript, or if this is not possible in a convincing way, to add a more explicit argument on why the model is still useful for the AEL question despite these deviations.

(3) Relatedly, if I understood correctly, the simulations began with all muscles set to zero excitation (Line 194). What were the initial muscle states (typically “activation” and either fiber length or force), and how were these states set? Please clarify this in the text. If the initial activations and forces were also zero, I think this is not realistic; even if excitations are realistically set to zero to begin the countermovement, the initial activations will be potentially non-zero from starting from an upright standing posture. It would be more realistic and potentially affect results/conclusions to start the muscle states at the states the hold the model statically in the initial pose, if this was not already done.

(4) Much of the setup and explanation of the findings emphasizes energy storage in tendons and its potential contribution to jump height. The Bobbert et al. (1996) study (already cited in this manuscript) argues I think fairly convincingly that the greater jump height in CMJ vs. squat jump is not primarily from greater elastic energy storage/return but rather from simply having more time to build up larger muscle forces. This makes sense given the mechanical requirements of jumping at the whole-body level: greater COM height is achieved only by greater COM take-off velocity, which achieved only by greater vertical impulse during ground contact, and CMJ has more time to generate that impulse. The same requirement holds for AEL jumping: to increase jump height, the added mass must increase vertical impulse. I thought the Introduction would benefit from noting this and from explaining how it was expected that AEL would achieve this. I’m not saying I think it shouldn’t be expected (as stated earlier, I believe this *has* to be expected, mechanically), just that I think it would help to explain this. Relatedly, please clarify in the text what the duration of the simulated movement was (e.g. 1.0 seconds or whatever) and if this was the same for all simulations and was sufficient time to confirm the model reached its maximum height. It may be helpful to compute COM height instead and check if this produces the same results/conclusions.

(5) The shift in focus midway through the Introduction to the role of biarticular muscles was surprising and it was unclear to me how it related to the AEL question. It seemed disconnected from the AEL-focused narrative of the rest of the Introduction. This model did not seem to isolate the effect of biarticular muscles, since it also removed the uniarticular hip muscle, and even if it accomplished this isolation, it was not clear to me what that information was adding to the AEL question. For brevity and clarity of focus, I suggest removing this particular model/condition and focusing on the 4-muscle model.

(6) For the AEL simulations, my understanding was that first one simulation was performed, optimizing the cost function of maximum jump-height with the added mass, then a mid-movement state from this solution corresponding to the assumed “switch” point between eccentric and concentric phases of ground contact, was extracted and used as the initial state for a second simulation/optimization with the added mass removed. I had a couple concerns with this approach. First, it assumes the “eccentric” portion of the jump is the same for both a jump executed to completion with added mass and for a jump executed with the added mass removed at the switch point. Is this assumption supported by evidence in the literature, and if not, could this be a reason why the simulations produced unexpected results? Second, it would help to explain in the text why this approach was used, e.g. where there technical restrictions? Was it not possible to simulate a case more similar to the human experiments where the added mass is suddenly removed mid-movement, involving only one simulation/optimization in the AEL condition? This approach seems more attractive because it doesn’t require the assumption I mentioned previously. If suddenly changing the mass causes discontinuity issues, a smoothed approximation may work, e.g. addedAELmass = a smoothed approximation of an IF statement checking the eccentric/concentric phase of the movement. The tanh function is common for this, for example:

IF (x <= d), f(x) = f1, ELSE f(x) = f2

can be smoothly approximated as:

h = 0.5 + 0.5*tanh(b*(x – d))
f(x) = f1 + h*(f2 – f1)

where b is the smoothing constant, larger b for less smoothing.

(7) Relatedly, can it be explained why the knee extensor fiber kinematics are a good choice to deciding the “Switch” point of the jump? I thought it would make more sense to use an identifier consistent with the jumping biomechanics literature which I think is typically not based on fiber kinematics.

(8) Line 122: some typos here (expecially and semgents)

(9) Line 237: where on the model was the AEL mass added?

·

Basic reporting

Major comment:
Results: I would have liked to see a simulation result to be able to judge how realistic the simulations were. I think it would have helped my understanding of the results in general, especially the relative differences that were mentioned in the section about squat depth and table 3.

Minor comments:
Line 113 – 126: this paragraph was not extremely well connected to the introduction. I would recommend explaining more clearly why the function of biarticular muscles is especially relevant to the problem of AEL loading
Line 203: The title “Cost function optimization” implies optimization of the cost function, therefore I would suggest slightly renaming this section.
Line 221-222: I would recommend to write an actual mathematical equation here
Figures 4 and 5: please increase font size of the axes of smaller figures

Experimental design

Major comment:
Line 232-248: did you test if you get the same result when you perform this two-step approach for the system without AEL? Based on the description of using the same states, I agree that theoretically the solution should be the same, but I know from experience that OpenSim can sometimes do something unexpected when used for simulations, while also the optimization could end up in local minima.

Validity of the findings

To contribute to the discussion, I wonder how much the timing of the removal of the AEL loading matters, which does not seem to be investigated. When looking at the experimental studies, it seems that the method used to remove the AEL loading was not as exact, and I wonder if the timing of this removal could matter as well.
Secondly, I also think that the joint kinematics could be different, despite the squat depth being the same (line 427-429). In practice, there are three joints (hip, knee and ankle) and two free parameters (x,y) hip position at the deepest point of the squat). Therefore, especially when considering only squat depth, the system is overdetermined and the same squat depth could be achieved with different joint configurations.

Minor comments:
Fig 2: Why does activation decrease below zero before the excitation onset?
Line 437-438: I felt that this statement of a changing activation pattern contradicts the statement in line 389-390 that no differences in EMG were found when an AEL was added.
Line 455-456: I found this statement a bit too strong and would add “in simulation” somewhere in the sentence

Additional comments

Regarding the attached code, I tried running “peerj-72346-Simulation_code\Simulation_code\two_muscle_model\BW\ Two_Segment_scaled_model_Optimization” and had to make the following adjustments, and was unsuccesful:
• Commented line 5 (“cd …”) in Two_Segment_scaled_model_FD. The folder name is different for each person. You could probably use the function “pwd” to make this work independently from the computer/user. For me it worked to comment it because I was in the correct folder anyways
• Change line 82 in the same folder to: movefile('Two_Segment_*', newdir);
• Change line 85 to cd 'Optimized_for_J_Results'
• Then I had to copy back the code into the original folder. I would not recommend to move code like that while it is running without a readme for the user explaining what happens..
• Then, I was unable to solve the next error in line 87 because the .sto file that should be imported is missing

---

## Round 0.2 · Minor Revisions

Apologies for the delays in obtaining reviews, but 2 reviewers have re-checked the resubmitted manuscript and have comments needing some moderate revisions, in particular regarding how "validation" is dealt with. I will check how this is handled in the newly revised manuscript. Thank you.

Reviewer 1 ·

Basic reporting

Please see "Additional comments" below

Experimental design

Please see "Additional comments" below

Validity of the findings

Please see "Additional comments" below

Additional comments

The "reviewing PDF" and the "resubmitted manuscript with changes tracked" had different line numbers. For example, the Methods section starts on line 133 in the PDF but line 147 in the tracked changes document. Neither set of lines numbers appeared to be consistent with the line numbers referenced in the rebuttal letter. For example, concerning the model's validity, the rebuttal letter referenced lines 386-398 but neither the PDF or the tracked changes document appeared to have any text added near those lines numbers that spoke to the topic from this part of the rebuttal. This made the revision/rebuttal difficult to evaluate and review.

I have a remaining concern on the demonstration of the model's validity. The other reviewer also appeared to have a similar concern. The rebuttal letter states "we have now also explained how our model is still “realistic” for its purpose throughout the manuscript. For example, ..." Respectfully, I did not see where this was done convincingly (see previous comment on the line numbers). It is also difficult to evaluate this kind of revision ("For example...") without being told as a reviewer specifically where these changes were made.

The validity of a new model and novel approach needs to be demonstrated before moving on to using the model for the research question. Doing this concurrently or after addressing the question is less effective. In response to the other reviewer, it looked like some data that could gauge the model's validity were added but were relegated to supplementary material. Especially for a journal like PeerJ that to my knowledge does not have strict limits on text length and figure count, I don't see a good reason to relegate this material to a supplement. It would be more effective and instill greater confidence in the reader if the model's validity is gauged as a distinct section of the methods/approach before moving on to addressing the purpose of the study.

·

Basic reporting

1. Line 85-97: in this added text, often the phrase “the impulse X should be greater” is used without specifying correctly what impulse this impulse should be greater than, because this should be an impulse, not e.g. “non-AEL CMJ”. Please rephrase these lines.

2. Please add equation numbers. In the equation in line 215, an x implies a cross product, while I think that a simple multiplication is meant. Therefore, I would suggest removing the x.

3. I still cannot read the axis numbers on the right side of figure 4 and 5 well, though this also has to do with the figure quality in the manuscript I reviewed. I would still suggest to further increase the font size.

4. Line 400: I think this should be "optimal solution" instead of "optimum solution"

5. Line 446-448: I agree with the response to the original comment about this phrase. However, it is not clear in the text here that you mean the model here, since the paragraph also discusses experiments. Therefore, I would clarify that in the manuscript as well.

Original comment:
Secondly, I also think that the joint kinematics could be different, despite the squat depth being the same (line 427-429). In practice, there are three joints (hip, knee and ankle) and two free parameters (x,y) hip position at the deepest point of the squat). Therefore, especially when considering only squat depth, the system is overdetermined and the same squat depth could be achieved with different joint configurations.
Author’s response: In reality, this might be the case. However, as stated in response to Reviewer 1, the model is highly constrained in order to explicitly test the assumption that improved performance can come from more elastic storage of energy and subsequently increase vertical impulse. As such, in our model, where horizontal movement of the hip and HAT segment are constrained, the same squat depth can only be achieved with the same joint kinematics, as there is only one degree of freedom which controls this (knee angle).

Experimental design

Good, no comments

Validity of the findings

Good, no comments

Additional comments

No further comments.

---

## Round 0.3 · accepted · Accept

I have checked the revised manuscript and approve that the revisions are adequate. Congratulations!